# Simultaneous Micro-Structuring and Surface Smoothing of Additive Manufactured Parts Using DLIP Technique and Its Influence on the Wetting Behaviour

**DOI:** 10.3390/ma14102563

**Published:** 2021-05-14

**Authors:** Florian Kuisat, Fabian Ränke, Fernando Lasagni, Andrés Fabián Lasagni

**Affiliations:** 1Institut für Fertigungstechnik, Technische Universität Dresden, George-Baehr-Str. 3c, 01069 Dresden, Germany; fabian.raenke@mailbox.tu-dresden.de (F.R.); andres_fabian.lasagni@tu-dresden.de (A.F.L.); 2CATEC Advanced Center for Aerospace Technologies, C/Wilbur y Orville Wright 19, 41309 La Rinconada, Seville, Spain; flasagni@catec.aero; 3Fraunhofer-Institut für Werkstoff- und Strahltechnik (IWS), Winterbergstr 28, 01277 Dresden, Germany

**Keywords:** nanosecond pulsed direct laser interference patterning, surface engineering, AM components, Scalmalloy^®^, Titanium 64

## Abstract

It is well known that the surface topography of a part can affect its function as well as its mechanical performance. In this context, we report on the surface modification of additive manufactured components made of Titanium 64 and Scalmalloy^®^, using Direct Laser Interference Patterning technique. In our experiments, a nanosecond-pulsed near-infrared laser source with a pulse duration of 10 ns was used. By varying the process parameters, periodic structures with different depths and associated roughness values are produced. Additionally, the influence of the resultant morphological characteristics on the wettability behaviour of the fabricated textures is investigated by means of contact angle measurements. The results demonstrated a reduction of the surface roughness of the additive manufactured parts (in the order of some tens of micrometres) and simultaneously the production of well-defined micro-patterns (in the micrometre range), which allow the wettability of the surfaces from 26° and 16° up to 93° and 131° to be tuned for Titanium 6Al 4V and Al-Mg-Sc (Scalmalloy^®^), respectively.

## 1. Introduction

Aluminium and titanium alloys are the most frequently used in the aerospace industry due to their lightness and their large specific mechanical properties (e.g., strength-to-weight ratio). The relative weight distribution belonging to the airframe of an aircraft is dominated by aluminium, generally with more than 60% wt. of the aerostructure, but also composite materials such as CFRP (Carbon Fibre-Reinforced Polymers) [1]. In the case of larger service temperatures or even when a greater corrosion resistance is required, Titanium alloys are mainly used up to 5–6% wt. in the airframe of commercial or sub-sonic military aircraft.

Additive Manufacturing (AM) technologies have shown a great potential for the development of light parts, due to the possibilities in topology optimization of the design [2,3]. In this sense, Titanium 64 has shown a good processability, achieving similar mechanical and corrosion resistance properties than the conventional aerospace version. On the other hand, the manufacturability of traditionally 2XXX, 6XXX or 7XXX aluminium alloys by AM powder bed fusion (PBF) -based processes is more complex, and therefore other alloy compositions such as Al-Mg-Sc (or Scalmalloy^®^) have been developed [4]. Additive Manufacturing technologies enable a simple layer by layer fabrication of complex 3D objects directly from CAD models. A number of AM technologies, such as Laser or Electron Beam Powder Bed Fusion (PBF-LB or PBF-EB), have in common that the powder particles are locally melted and solidified [5,6]. Due to the characteristic of the manufacturing process, the fabricated components have a relatively high initial roughness level on their surface, which is typically >8 µm (Sa) for PBF-LB and could affect their mechanical performance [7]. For instance, the mechanical behaviour (e.g., fatigue life) of the AM-parts can be improved by increasing the surface quality (e.g., reducing surface roughness) as already discussed by Spierings et al. and Sakar et al. [8,9]. In other words, the final quality of additive manufactured surfaces is a well-known problematic, that needs to be addressed.

Many ongoing research works concentrate on the optimization of the AM process or the surface finishing techniques of the as-built components to improve its surface quality [10,11,12,13]. For instance, Tyagi et al., demonstrated an improvement of the surface quality by optimizing the manufacturing processes and processing parameters, which are often limited by physical or material properties [13]. To subsequently improve the surface quality of AM components, post-treatment processes are also frequently necessary. Common finishing techniques are milling, blasting or vibration grinding [14,15]. In addition, methods such Hot Isostatic Pressing (HIP) are effective to reduce the number of defects or alter the microstructure of AM manufactured components in order to improve their mechanical properties [16,17]. Other options to modify the surface topography of AM parts can be achieved by the use of laser radiation [18,19,20]. For instance, in laser polishing, the energy of the laser radiation is applied to the surface to melt the roughness peaks, which leads to a smoothing effect of the initial topography as a result of the surface tension, where the molten material can flow into the valleys. The topographical characteristics of the smoothed topography depend on the kind of operation (continuous or pulsed wave modes), the process parameters as well as the material properties [21]. For instance, Marimuthu et al. have investigated that continuous wave laser polishing of AM surfaces can be a useful method to improve the surface quality of Ti64 [22], whereas Perry et al. demonstrated a roughness reduction by more than 60% using a nanosecond pulsed laser source [23]. Many other reports on laser polishing have been published by other researchers [24,25,26]. Most of these post-treatment processes have in common that they aim to enhance the surface quality (e.g., roughness reduction or reduction of defects), which is certainly a very critical aspect of these components for numerous applications. However, the quality is not the only aspect that needs to be considered. In addition to the improvement of the surface quality of parts, laser-based micro-structuring methods have also shown to be able to provide surfaces with specific surface properties, which cannot be addressed by laser polishing. Within these methods, Direct Laser Writing (DLW) and Direct Laser Interference Patterning (DLIP) have shown to be effective techniques for fabricating microstructures with different feature sizes and producing functional surfaces. In the case of DLW, typical feature sizes are typically in the range of 10 µm to 50 µm [27], while the DLIP technology can produce elements with feature sizes between 0.5 µm and 20 µm (even bellow the diffraction limit) [28]. In the past, numerous research works have been performed for patterning different materials using these methods and improve their technical functions as a result of the produced microstructures. For example, Milles et al. and Huerta-Murillo et al. demonstrated (super-)hydrophobic surface conditions after the fabrication of structures using DLW and DLIP processes on pure aluminium and titanium alloy, respectively [29,30]. Other applications have been described for instance by Rosenkranz et al. and Lu et al. who shown the possibility to control the wear and corrosion resistance of surfaces, respectively [31,32]. However, to the best of our knowledge, laser-based fabrication methods have not been combined so far, allowing both an improvement of the surface quality (reduction of general surface roughness) of AM-parts and simultaneously providing them with hydrophobic properties by creating well-defined periodic structures.

In this frame, the presented work examines the utilization of Direct Laser interference Patterning technique as an innovative method for reducing the surface roughness of AM-components and simultaneously produce a structured surface on Titanium 64 and Scalmalloy^®^ alloys to control their wettability. Important aspects of the investigation are the determination of the achievable roughness level as well as the topographical characteristics of the produced patterns as function of the process parameters. Finally, the optimal parameters to reach higher water contact angles are discussed. The appropriate combination of roughness reduction and surface micro-structuring in a one-step process is expected to improve the industrial applicability of AM components and enable the integration of different functions.

## 2. Materials and Methods

### 2.1. Materials

Substrates of Ti64 (Titanium 6Al4V, Suppl. Renishaw plc, Wotton-under-Edge, UK) and Scalmalloy^®^ (Al-based alloy, Suppl. Toyal Europe, Accous, France) utilized in this contribution. The specimens were manufactured by laser-Powder Bed Fusion (L-PBF) on a Renishaw AM250 (Ti64) and RenAM 500 (Scalmalloy^®^) systems (Wotton-under-Edge, UK). PBF-LB belongs to the group of additive manufacturing processes and provides the highest resolution and accuracy of today’s AM techniques for metals. Both materials are characterized by high resistance and ductility combined with lightweight and are today two of the most popular metallic materials for space-flight and aircraft components [33,34]. Further information about the used materials as well as the used manufacturing process have already been published elsewhere [20]. Based on the used fabrication process, the initial maximum height (Sz) as roughness parameter was determined to be 94.1 ± 22.9 µm for Ti64 and 83.1 ± 21.8 µm for Scalmalloy^®^, respectively. Before the laser experiments, the specimens were cleaned with an alcohol-based agent (isopropanol) to remove dirt particles.

### 2.2. Direct Laser Interference Patterning

The additively manufactured samples were processed using the DLIP technology. To produce line-like patterns, a laser beam is split into two separate beams which were overlapped on the substrate at a certain angle of intersection *θ* (see Figure 1a)

At this region, a periodic intensity profile is created and its shape can be directly transferred to the material surface by selective melting or ablation at the interference maxima positions [35]. The DLIP workstation uses a slab-type solid-state laser (Edgewave IS400) (Würselen, Germany), which emits light with a fundamental wavelength (*λ*) of 1064 nm with 10 ns pulse duration (*τ*) and a maximum output power (*P_L_*) of 200 W. The laser fluence (*F*) or laser density per pulse was set to 3.2 J/cm^2^ and the spot size of the interference area was about 0.1 × 5 mm^2^ (irradiated spot dimension *d_y_* and *d_x_* on the sample). The long-shaped laser spot permits high processing rates by enabling the patterning of larger areas per laser pulse (for further information see [36]). Figure 1b shows schematically the scanning strategy utilized in this work. By setting the pulse overlap (*OV*), and thus the number of pulses per area, it is possible to control the energy introduced into the material. In this study, the pulse overlap between two individual laser spots was varied from 0% to 98%. For instance, 0% overlap denotes that only one pulse reaches the same position on the material surface, while 98% denotes that 50 pulses have been used. By translating the samples in x- and y- directions, using a linear axis system, larger areas can be textured. This have been performed using an Aerotech XY-moving stage, allowing translational speeds up to 1 m/s.

In this work, the spatial period (*Λ*) of the interference pattern was set to 21 µm, which corresponds to intersection angle (*θ*) of 2.9°. The laser experiments were conducted in air, at normal conditions of pressure (1013 hPa) and temperature (22 °C). Further information about the setup can be found elsewhere [36].

### 2.3. Surface Characterization Methods

For the analysis of the surface morphology, a confocal microscope (CM, Sensofar S-Neox) (Terassa, Spain) with 20× and 50× magnification objectives was used. The roughness parameter Sz (maximum height), which is an extension of Rz to a surface, was measured according to DIN EN ISO 25, 178 [37]. This parameter is defined as the sum of the height value of the highest peak and the height value of the strongest pit within the definition area. Sz has been selected as parameter of investigation because it represents better the maximal differences of the topographical elements (height differences) and, thus, giving a better description of the influence of laser treatment on the surface topography of the AM parts. For statistical analyses, the average of six measurements (at different positions) were taken. Additionally, a Scanning Electron Microscope (SEM, ZEISS Supra 40 VP) (Jena, Germany) was utilized to analyse the surface topography of the samples at an acceleration voltage of 5.0 kV. The wettability characterization in terms of the static water contact angle (WCA) was performed using a contact angle measurement system from Krüss GmbH (DSA 100 S) (Hamburg, Germany). Each measurement was performed at least five times with a droplet volume of 2 µL for statistical analysis at ambient conditions (22 °C and 16% R.H.). The standard deviation of the measurements describes the measurement error.

## 3. Surface Modification by DLIP Technique

In a first set of experiments, the influence of the pulse overlap (number of pulses per area) on both surface roughness and depth of the line-like textures was evaluated. As mentioned before, the objective is to reduce the general level of the surface roughness and simultaneously to control the wettability of the treated surface due to the created periodic structure using two-beam DLIP method with nanosecond pulses. The laser parameters for producing these patterns were selected according to previous investigations, which pointed out that high laser fluences (3.2 J/cm^2^) lead to smoother surfaces and higher structure depths (not shown).

The created DLIP pattern consists of line-like structures with a periodicity of 21.0 µm, which were generated over a total area of 5 × 10 mm^2^. Figure 2 shows representative SEM images taken before and after the DLIP process of Ti64 (a–c) and Scalmalloy^®^ (d–f). In detail, the images represent the initial surfaces (a and d) and selected line-like patterns produced with a pulse overlap of 50% (b and e) and 98% (c and f), respectively. The SEM images show that the untreated surfaces have an overall undefined and random morphology with large material particles, which can be attributed to different aspects (e.g., powder properties or manufacturing process) and has been reported in previous publications [38,39]. Especially, the Ti64 surface (a) is characterized to have roughness peaks, which are typical for unmodified additive manufactured components [40]. The large roughness level mainly induced by the locally melted powder particles sticked to the surfaces. In contrast, due to the used DLIP laser treatment with ns laser pulses, periodic structures with a line-like geometry are observed on all laser modified samples. As it is already known, in the case of ns pulses, the irradiated regions at the positions corresponding to the interference maxima are molten and ablated [41,42]. On the other hand, it is also well known that nanosecond pulses produce a relatively large heat affected zone and thus allowing to create a pool of molten metal [20,43]. Both effects are indispensable for treating the rough additive manufactured components made of Ti64 and Scalmalloy^®^. In case of Ti64, it can be seen that the surface of the laser-treated parts with different parameters (Figure 2b,c) have an overall modified topography with periodic features combined with the characteristic initial surface roughness in the background (Figure 2a). For example, Figure 2b shows a surface with considerable initial roughness combined with a poorly defined DLIP structure (treated at 3.2 J/cm^2^ with 50% overlap), which indicates that the surface was not significantly affected by to the laser radiation. In contrast, Figure 2c shows the treated Ti64 surface with a clear visible line-like texture, since in this case a high overlap was used (treated at 3.2 J/cm^2^ with 98% overlap). The expected period of the line-like textures of 21.0 µm is indicated in the insert image in Figure 2c. However, the image also demonstrates that the texture is not perfectly formed, which is denoted by the cavities of the structure. An explanation for the mentioned discontinuities can be related to the locally adhered material particles with different sizes which are not distributed homogeneously over the surface.

In contrast, the surface morphologies of the DLIP-treated Scalmalloy^®^ samples (Figure 2e,f) are characterized by more uniform and distinct textures. In this case, the initial surface roughness can be slightly distinguished in the background. Higher pulse overlaps result in a stronger interaction between the laser radiation and the substrate, modifying the surface more effectively as can be seen in Figure 2f. The textures also show a spatial period of 21.0 µm, which is in agreement with the used intercepting angle of 2.9° (see insert in Figure 2f).

Figure 3 shows exemplarily 3D topography images of the untreated and DLIP treated samples measured using confocal microscopy for Ti64 (Figure 3a–c) and Scalmalloy^®^ (Figure 3d–f), respectively. The images correspond to samples treated with overlaps OV of 50% and 98%. It can be seen on both reference samples, that the topography is dominated by large valleys and hills, as has been discussed before. The Ti64 sample (a) is characterized by drop-shaped material particles, while the Scalmalloy^®^ sample (d) has fewer drop-shaped particles but also large elevations. Such differences between the initial surface morphology can be attributed to the various material properties such as particle size or material element distributions as well as the manufacturing processes.

The images of the DLIP treated surfaces show that, for instance, for an overlap (OV) of 50% the initial rough topography is dominant for both materials. The recorded surface topography looks nearly similar to the initial surface, which can be explained by the insufficient cumulated energy applied to the material (for OV = 50%, two laser pulses hit the same position during the laser treatment). In contrast, for the higher pulse overlaps (OV = 98%), the DLIP structure is clearly visible, as shown in Figure 3c,f. However, in the case of the Ti64 material (Figure 3c), the DLIP structure is less dominant compared to Scalmalloy^®^ (Figure 3f). This effect can be explained by two reasons. Firstly, the Ti64 alloy has a significantly higher initial surface roughness, and as it is known from other studies, higher roughness has a negative influence in the formation of periodic patters in DLIP [44,45]. The second reason can be attributed to the thermal properties of the used alloys such as the thermal conductivity. For instance, the thermal conductivity of pure titanium and aluminium vary from 22 to 19 W/m⸱K and 236 to 232 W/m⸱K, between 273 and 600 K [46], showing a difference of one order of magnitude. Thus, the higher thermal conductivity of Scalmalloy^®^ leads to a larger volume of material that is molten at the interference maxima positions which contributes to the formation of the periodic structure. Furthermore, in case of DLIP treatment with ns pulses, the contribution of the molten material on the structure formation has already been discussed, driven mainly by Marangoni flow and recoil pressure for low and high intensity levels, respectively [47,48].

After these first impressions of the DLIP modified surfaces, a systematic analysis was carried out to evaluate the capability of both reducing the initial surface roughness as well as to produce periodic surface structure with the main objective of controlling the wettability of the AM parts. The results in terms of the variation of the pulse overlap on the surface roughness and structure depth are summarized in Figure 4a,b for Ti64 (black triangles) and Scalmalloy^®^ (red circles). For Sa roughness values, see Figure A1 in Appendix A. For a better understanding of the effectivity of the laser process to control the general surface roughness (Figure 4a), the initial roughness value is highlighted in green and purple for Ti64 and Scalmalloy^®^, respectively. The maximum heights (Sz) were observed for the samples patterned at lowest pulse overlaps (0–50%). By increasing the pulse overlap, the surface roughness decreased continuously. However, some differences have been observed depending on the used material. In case of the titanium alloy, the Sz roughness was reduced from 94.1 ± 23.3 µm to 67.7 ± 7.2 µm, reaching the lowest value for the highest used pulse overlap (OV = 98%). For the aluminium-based material, the Sz roughness was reduced from 83.1 ± 21.8 µm to 47.3 ± 3.6 µm, which corresponds to a roughness reduction of 43%. In the last case, increasing the pulse overlap from 90% to 98% did not shown any significant improvement of the surface quality.

Another important observation besides the reduction of the surface roughness is that the measured standard deviation for Sz also significantly decreased, which indicates a better homogeneity as well as a more uniform morphology. This effect can be explained by the fact that the number of pulses per area increases continuously with an increased overlap and, thus, a higher amount of material is molten and selectively ablated. This indicates that the roughness hills have been strongly remelted and even partially ablated with each laser pulse.

In turn, the diagram in Figure 4b shows the evolution of the structure depth of the interference texture also as function of the pulse overlap. As expected, the measured depth increases with the pulse overlap, especially in the range from 80% to 98%. In particular, this range corresponds to a pulse number of 5 and 50, and thus resulting in cumulative fluences of 16 J/cm**^2^** and 160 J/cm**^2^**, respectively. In particular, it can be observed that the structure depths for Scalmalloy**^®^** starts to growth significantly, above a pulse overlap of 80%. For instance, a maximal structure height of 11.4 ± 2.8 µm was achieved for Scalmalloy**^®^** at a pulse overlap of 98%. For Ti64, overlaps over 90% (corresponding to 10 laser pulses and a cumulative fluence of 32 J/cm**^2^**) were necessary to produce a well-defined DLIP geometry. This effect could be related, as mentioned before, to the higher initial roughness for this material. The maximal structure depth obtained was 3.9 ± 0.6 µm also at 98% of overlap.

Using this information, the aspect ratio of the created textures can be calculated, which is defined as the quotient between the interference pattern depth and the spatial period. Here, aspect ratios of 0.18 and 0.54 can be calculated for Ti64 and Scalmalloy**^®^** materials, respectively. It should also be mentioned that deeper structures are accompanied by larger deviations (see error bars in Figure 4b), which was observed especially in the case of Scalmalloy**^®^**.

## 4. Characterization of the Wettability Properties

The evolution of the wetting properties of the DLIP treated surfaces was studied by static water contact angle (WCA) measurements every 7 days during a period of about 49 days. Figure 5 shows the WCA temporal evolution of the untreated reference samples (dashed lines) as well as representative DLIP-treated samples (coloured symbols) for (Figure 5a) Ti64 and (Figure 5b) Scalmalloy^®^ depending on the used pulse overlap. Both diagrams also include the droplet shape of the initial surface and of the surface with the highest WCA to visualize the changes that have occurred.

As can be seen from the plots, all surfaces exhibit hydrophilic behaviour immediately after DLIP treatment. The averaged WCAs of the untreated samples were 25.9 ± 5.8° and 16.3 ± 3.1° for Ti64 and Scalmalloy^®^, respectively. However, upon closer consideration of the contact angle, all laser patterned samples show a significant increase in the WCA over time. In the case of Ti64 (Figure 5a), the three showed curves, which represents different pulse overlaps, show a comparable evolution during the first 50 days. The measured WCAs ranges from 32.7° to 55.7° immediately after the laser process and increases over time to contact angles between 87.4° and 93.4°, which is close to the hydrophobic boundary condition (WCA ≥ 90°). The WCA increase as function of time can be attributed to the change in surface chemistry, which is produced after the laser treatment as has been shown by several authors [49,50,51]. In particular, changes in the amount of the organic compounds (e.g., carbon content) on the material surface take place, and thus the non-polar character of the surfaces is increased. The highest measured WCA was observed for the AM samples processed with 90% pulse overlap, where the contact angle increases from 32.7 ± 8.1° to 93.4 ± 2.7°, which represents a percentage increase of approximately 65% in WCA. However, the other evaluated process conditions produced samples with very similar values. The reason for this behaviour can be explained by the insufficient modification of the topography in all cases. For instance, the roughness value ranges between 95.2 µm to 67.7 µm and the interference pattern ranges from 0.1 to 3.9 µm for pulse overlaps between 50% and 98%, respectively. It can be assumed that the background roughness (which were presented in Figure 3a–c) dominates the WCA behaviour and further increase in WCA requires a lower surface roughness combined with a stronger periodical texture depth.

The Scalmalloy^®^ specimens (Figure 5b) treated by DLIP were highly hydrophilic directly after the laser process, reaching WCAs between 5.1° and 14.3°. Like for the Ti64 samples, the WCA increased over time up to 131.4°. However, the evolution of the contact angles indicates important differences depending on the used overlap. The maximum recorded WCAs after 50 days were 81.3 ± 5.0°, 107.9 ± 7.6° and 131.4 ± 1.9° for pulse overlaps of 50%, 90% and 98%, respectively. Such differences of the WCA can be correlated with the different depths of the interference structures, being 0.3, 4.5 and 11.4 µm for the pulse overlaps of 50%, 90% and 98%, respectively.

In other words, higher depths of the interference structures result in higher contact angles, which means that the created texture importantly affect the WCA. Here, the periodic textures can provide air pockets between the material and the water droplets, resulting in a composite surface according to Cassie and Baxter [52]. It is important to mention that, although the general surface roughness was reduced by the DLIP treatment, especially when large overlaps were used, an increase in the WCA was observed denoting the importance of producing a well-defined and ordered surface topography for controlling the surface wettability. Overall, it can be summarized that, in the best case, the contact angles are about 4 and 8 times higher after the DLIP treatment for Ti64 and Scalmalloy^®^, respectively, compared to the untreated reference samples.

## 5. Conclusions

The surface morphology and related wetting characteristics of the periodic line-like textures on additive manufactured titanium and aluminium alloy by DLIP technique were systematically investigated. The following important statements can be derived from this research work:
(i)Firstly, the feasibility of the ns DLIP treatment for reducing the initial roughness values of additively manufactured Ti64 and Scalmalloy^®^ specimens was demonstrated.(ii)In particular, the DLIP process permitted the reduction of the surface roughness from 94.1 ± 23.3 µm to 67.7 ± 7.2 µm for Ti64 and from 83.1 ± 21.8 µm to 47.3 ± 3.6 µm for Scalmalloy^®^, representing and improvement of 28% and 43% for both materials, respectively.(iii)It was also possible to prove an improvement of the roughness homogeneity, what has been demonstrated by the reduced standard deviation of the measured surface roughness Sz.(iv)Beside the reduction of the initial surface roughness, it was also shown the ability to produce line-like patterns with structure depths up to 3.9 ± 0.6 µm and 11.4 ± 2.8 µm for Ti64 and Scalmalloy^®^, respectively.(v)Finally, it was shown that the DLIP treatment allows increasing the contact angle for both materials. In particular, the static water contact angle increased from 25.9° to 93.4° and from 16.3° to 131.4° for Ti64 and Scalmalloy^®^ alloys, respectively.

In summary, the DLIP treatment of AM-parts showed a clear modification of the surface topography and roughness as well as significant increase in the water contact angle, which can be of advantage for some applications in aerospace industry and extend the applicability of additive manufactured components beyond the current state of the art.

## Figures and Tables

**Figure 1 materials-14-02563-f001:**
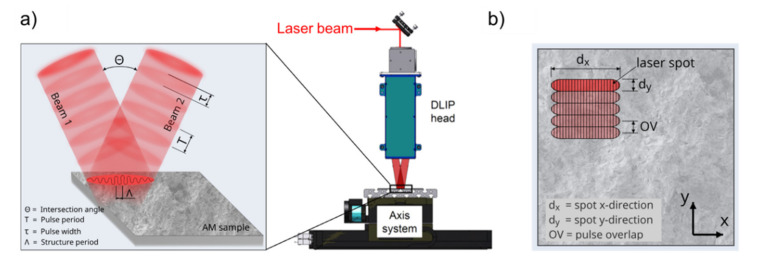
(**a**) Experimental setup (TU Dresden, Fraunhofer IWS) developed for processing large area surfaces with an illustration of two elongated laser beams overlapping on the substrate surface (left side); (**b**) Schematic drawing of the moving process strategy for the DLIP process with the most relevant parameters.

**Figure 2 materials-14-02563-f002:**
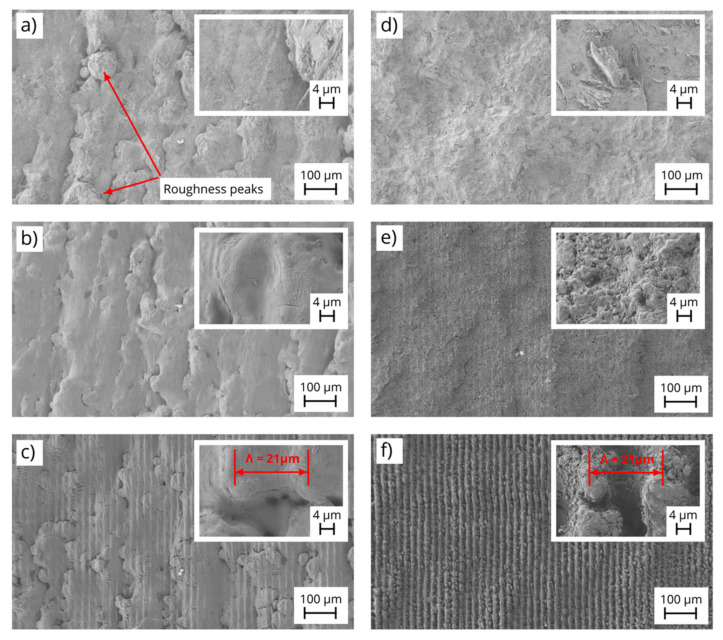
SEM images of initial (**a**,**d**) and DLIP-treated samples using pulse overlaps of 50% (**b**,**e**) and 98% (**c**,**f**) and a set laser fluence of 3.2 J/cm^2^ for AM specimens made of Ti64 (**a**–**c**) and Scalmalloy^®^ (**d**–**f**).

**Figure 3 materials-14-02563-f003:**
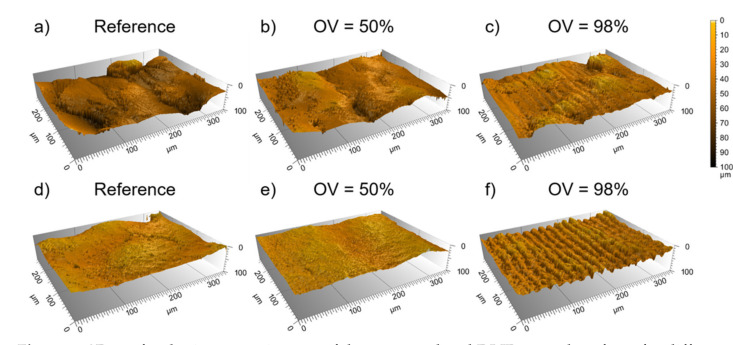
3D confocal microscope images of the untreated and DLIP treated surfaces for different pulse overlaps (OV = 50% and 98%) for Ti64 (**a**–**c**) and Scalmalloy^®^ (**d**–**f**) specimens.

**Figure 4 materials-14-02563-f004:**
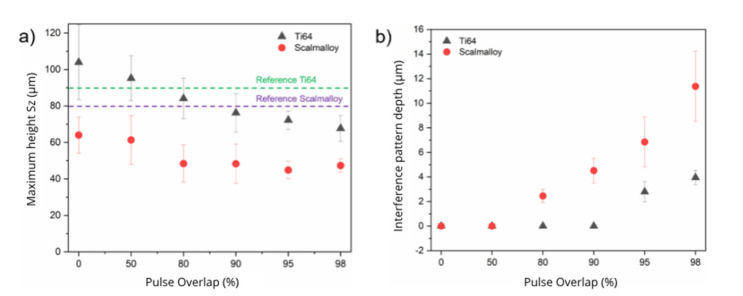
(**a**) Surface roughness Sz and (**b**) interference pattern depth as a function of the pulse overlap for the DLIP-treated titanium and aluminium alloys. The used laser fluence was 3.2 J/cm^2^ for all experiments.

**Figure 5 materials-14-02563-f005:**
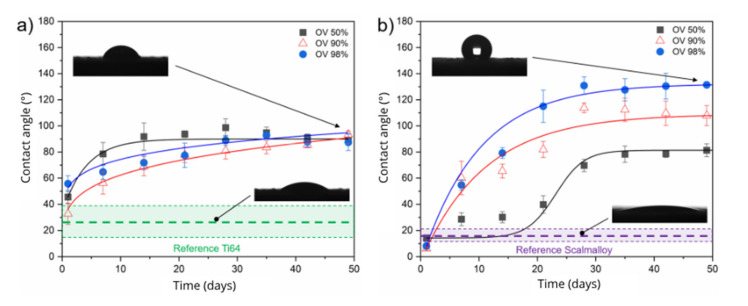
Temporal trend of static water contact angles (WCA) for DLIP modified (**a**) Ti64 and (**b**) Scalmalloy^®^ specimens depending on the pulse overlap (OV = 50%, 90% and 98%).

## Data Availability

The data presented in this study are available on request from the corresponding author.

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
