# Peer review of "Simultaneous Micro-Structuring and Surface Smoothing of Additive Manufactured Parts Using DLIP Technique and Its Influence on the Wetting Behaviour"

_materials, 2021, doi:10.3390/ma14102563_

Round 1

Reviewer 1 Report

This paper calls: "Simultaneous micro-structuring and surface smoothing of additive manufactured parts using DLIP technique and its influence on the wetting behaviour" and concerned of treatment on Titanium 64 and Scalmalloy by Direct Laser Interference Patterning technique (DLIP). The main advantage of this work is clear experimental material presentation, parameters of treatment (laser power, pulses duration) and interesting results in changing of surface properties and increase of surface's wettability of such new materials as Titanium 64 and Scalmalloy.

In my opinion this work must be presented in present form.

Reviewer 2 Report

Review for the “Simultaneous micro-structuring and surface smoothing of additive manufactured parts using DLIP technique and its influence on the wetting behaviour”

The work is interesting and I think the research community will enjoy the results, however I have indicated some comments for further improvements as listed below:

The authors presented some post processing but there are plenty in literature which were not considered, therefore I strongly advice having a look : https://doi.org/10.3390/ma14030658  ; https://doi.org/10.1016/j.cirp.2019.04.116

Otherwise the introduction should be more critical for the past research, because there are no much elements of criticism rather only discussion what was made

Please elaborate more in scientific novelty of this work, which will attract more readers!

Is there any physical reason selecting “Sz” as parameter of investigation ?

In line 137 you provided some interesting set up details, they all are in line with [35]?? Otherwise I curios in which base were selected

I am not sure if I understand well in which environment were performed the laser ablation ?

A citation is required for the “EN ISO 25178.”

For statistical analyses” that’s very good approach, can you indicate how many measurements were conducted ? and if so there was used the average ?

“or unmodified additive manufactured components.” A citation is required

Can you indicate in which Figure were observed and if so please point out with an arrow these structures

My question about the statistical meaning indicate in Figure 4 is that if you measured the surface in Figure 3 that means you sample size seems very small and not sure if it is representative !

Not sure fi I understand clearly how you measured this “measurements during a period of about 50 days” was conducted measurements every days or just after 50day..please elaborate little in this aspect

Quite interesting this increase in contact angle after laser processing, even you pointed out a ref yet seems very curios

Reviewer 3 Report

The Authors focused on the surface modification of additive manufactured components made of Titanium 64 and Scalmalloy, using Direct Laser Interference Patterning technique. They applicate a nanosecond-pulsed near-infrared laser source with a pulse duration of 10 ns. The influence of the resultant morphological characteristics on the wettability behaviour of the fabricated textures has been investigated by means of contact angle measurements. The article seems to be interesting for a large number of scholars and engineers. It creates a logical scientific experimental research and that is why in my opinion could be published in "Materials". Some of the comments on the manuscript are listed below.

1) Line 24 and 25; some keywords have been already used in the title of the manuscript. Please change them into different ones (to avoid the keywords repetition with the words used in the title).

2) Line 29; what do the Authors mean by large specific mechanical properties?

3) Line 235; the thermal conductivity for metals is usually highly nonlinear relationship vs. temperature. Could you introduce some more information why do you mention a constant value of thermal conductivity in a wide range of temperature?

Round 2

Reviewer 2 Report

.